# Differential Response of MDA-MB-231 and MCF-7 Breast Cancer Cells to In Vitro Inhibition with CTLA-4 and PD-1 through Cancer-Immune Cells Modified Interactions

**DOI:** 10.3390/cells10082044

**Published:** 2021-08-10

**Authors:** Kamil Grubczak, Anna Kretowska-Grunwald, Dawid Groth, Izabela Poplawska, Andrzej Eljaszewicz, Lukasz Bolkun, Aleksandra Starosz, Jordan M. Holl, Marta Mysliwiec, Joanna Kruszewska, Marek Z. Wojtukiewicz, Marcin Moniuszko

**Affiliations:** 1Department of Regenerative Medicine and Immune Regulation, Medical University of Bialystok, Jerzego Waszyngtona 13, 15-269 Bialystok, Poland; annammkretowska@gmail.com (A.K.-G.); dawid.groth@umb.edu.pl (D.G.); andrzej.eljaszewicz@umb.edu.pl (A.E.); aleksandra.starosz@gmail.com (A.S.); jordanmholl@gmail.com (J.M.H.); 2Department of Medical Pathomorphology, Medical University of Bialystok, Jerzego Waszyngtona 13, 15-269 Bialystok, Poland; iptaborda@gmail.com; 3Department of Haematology, Medical University of Bialystok, M. Sklodowskiej-Curie 24A, 15-276 Bialystok, Poland; lbolkun@gmail.com; 4Department of Oncology, Medical University of Bialystok, Ogrodowa 12, 15-027 Bialystok, Poland; marta.mysl@gmail.com (M.M.); joannak@umb.edu (J.K.); mzwojtukiewicz@gmail.com (M.Z.W.); 5Department of Allergology and Internal Medicine, Medical University of Bialystok, M. Sklodowskiej-Curie 24A, 15-276 Bialystok, Poland

**Keywords:** immune checkpoint inhibitors, CTLA-4, PD-1, breast cancer, anti-tumor immunity

## Abstract

Drugs targeting immune checkpoint molecules have been found effective in melanoma, lung cancer, and other malignancies treatment. Recent studies on breast cancer demonstrated the significance of inhibitory anti-CTLA-4 and anti-PD-1 in the regulation of disease progression. However, seemingly the same types of breast cancer do not always respond unambiguously to immunotherapy. Thus, here we set out to analyze the in vitro effects of inhibiting CTLA-4 and PD-1 on interactions between co-cultured lymphocytes and two selected breast adenocarcinoma cell lines. Breast cancer cells were co-cultured with lymphocytes to evaluate the effects of CTLA-4 and PD-1 inhibition. Proliferation, cell cycle, and viability assessment were measured in cancer cells. IFN-gamma, IL-10, perforin, granzyme B production, and CTLA-4 and PD-1 expression were analyzed in lymphocytes. We found that administration of anti-CTLA-4 improved the anti-cancer activity of T cells with reduced proliferation and viability of MDA-MB-231. Lack of response was observed in the context of MCF-7. In addition, differential expression of checkpoint proteins was found between studied cancer cells lines. Inhibition of molecules was followed by IL-10 and IFN-gamma decrease in lymphocytes co-cultured with MDA-MB-231, not demonstrated in reference to MCF-7. Furthermore, CTLA-4 blockage was associated with reduction of CTLA-4+ and PD-1+ lymphocytes in MDA-MB-231, with a significant increase in MCF-7, reduced by anti-PD-1. Altogether, our study revealed that anti-CTLA-4 and anti-PD-1 treatment can improve lymphocytes effects on breast cancer cells. Favorable effects seemed to be related to breast cancer cells features as differential responses were reported. Novel blocking antibodies strategies should be tested for more effective cancer inhibition.

## 1. Introduction

Breast cancer is the most common cause of cancer-related death in women. Being the most frequently diagnosed cancer in females it accounts for nearly 23% of total cancer cases [1]. Molecular distinction seen in breast tumors engages numerous therapies to be used in the management of this disease. Aside from surgery, chemo- and radiotherapy, which serve as the basis of breast cancer treatment, targeted therapy with agents directed at specific molecule receptors seems to be taking the leading role in several cases [2]. An example of such is endocrine therapy, where a selective estrogen receptor modulator-tamoxifen is used against ER-positive breast cancer cells resulting in inhibition of their growth and apoptosis [3]. On the contrary, the triple-negative breast cancer (TNBC) subtype has the highest likelihood of recurrence and poorest survival prognosis [4]. The efficacy of current therapeutic solutions remains unsatisfactory and therefore there is a great demand for research into this field.

Immunotherapy has become a leading treatment option for patients with melanoma or lung cancer and is soon to be approved for kidney, bladder, and prostate cancer therapy [5,6]. Immune response checkpoint regulators including CTLA-4 and PD-1 have been shown to play a critical role in cancer development through interactions with B7 proteins, particularly CD80 and CD86. According to Xu et al., the B7 protein family was observed to be amplified in breast cancer, which allowed for the introduction of checkpoint protein modulation in breast cancer treatment [7]. In addition to B7 receptors, the upregulation of their ligands CTLA-4 and PD-1was noted [8,9]. In meta-analysis studies, CTLA-4 polymorphisms have been shown to correlate with breast cancer susceptibility, emphasizing the importance of CTLA-4 in regard to tumor development [10]. Recent research from clinical trials has demonstrated that a single dose of anti-CTLA-4 monoclonal antibody (ipilimumab) leads to increased intratumor T cell density [11], which is associated with a lower risk of chemotherapy resistance and higher overall survival in breast cancer patients [12,13]. In the context of the PD-1/PD-L1 axis, recent studies have demonstrated that breast cancer patients with low levels of PD-1+ tumor-infiltrating lymphocytes (TILs) and high expression of PD-L1 within tumors showed the worst survival rate. This fact is likely associated with tumor-induced suppression of the immune response as a consequence of the PD-L1 interaction with lymphocyte-expressed PD-1 [14]. Promising results have also been seen in PD-L1-positive TNBC patients phase Ib clinical trials where a single-agent pembrolizumab showed antitumor activity in 18.5% of subjects, achieving response durations from 15 to more than 47 weeks, with the best standard chemotherapy response duration oscillating within 4 to 12 weeks [15,16]. On the other hand, results of a phase 2 study KEYNOTE-08 evaluating pembrolizumab monotherapy in TNBC treatment proved somewhat modest, with an objective response rate of around 5.7% in the PD-L1-positive population [16].

An increasing number of studies also incorporate CTLA-4 or PD-1 inhibitors to be used as adjuvant therapy, thereby improving anti-cancer agent activity [17]. In experiments studying populations of mice, the application of anti-CTLA-4 antibodies was found to improve the antitumor activity of gemcitabine by leading to sustained, long-term tumor size reduction [18]. Concomitant treatment of tremelimumab (anti-CTLA-4 IgG_1_ antibody) with exemestane has been tested in hormone-responsive breast cancer, demonstrating the induction of ICOS+/CD4-/CD8+ lymphocytes, suggesting enhanced activation of conventional T cells in regard to antitumor immunity in addition to immune suppression as a consequence of reduced regulatory T cell numbers [19]. Sagit-Barfi et al. show that despite ineffective blockage of the PD-1/PD-L1 pathway with anti-PD-L1 alone, its combination with ibrutinib (molecule binding to Bruton’s tyrosine kinase) significantly enhanced the antitumor activity of T-cells in mice inoculated with breast and colon carcinoma, among others [20]. Similar results were obtained in mice treated with a combination of anti-PD-1 antibody and doxorubicin, which led to reduced metastasis of 4T1 mammary tumor, unlike treatment with anti-PD-1 alone [21]. Despite somewhat promising results of therapeutic immune checkpoint inhibitors, selection of appropriate agents likely depends on the type of treated cancer, as suggested in recent studies, where tumor cells of breast cancer 4T1 cell line demonstrated significantly lower levels of the CTLA-4 ligand (B7-1) compared to colon carcinoma CT26 cell line, despite similar lower level expression of PD-L1 [22]. The number of studies focusing on the assessment of CTLA-4/ PD-1 inhibition on interactions between lymphocytes and different subtypes of breast cancer remains scarce.

Although research into potential novel approaches to treatment in triple-negative breast cancer does not seem to raise any ambiguity due to its dismal prognosis [4], this cannot be said for ER-positive, PR positive, and HER2-negative breast cancer because these patients show a rather favorable outcome [23]. This tendency has however resulted in a limited number of studies focusing on new therapeutic solutions for treatment-resistant luminal A-type breast cancer patients. To date, the assessment of CTLA-4/ PD-1 inhibition on interactions between lymphocytes and different subtypes of breast cancer has not yet been thoroughly analyzed. Several studies have however associated poorer clinical outcomes of breast cancer patients with the expression of CTLA-4 in the tumor microenvironment, justifying the enormous need for further research in this field [24]. Saleh et al. found that MDA-MB-231 (TNBC) cells upregulated CTLA-4 expression in CD4+CD25− T lymphocytes and downregulated PD-1 expression in those CD4+CD25+. On the contrary, the ER-positive breast cancer cells (MCF-7 cell line) greatly upregulated the PD-1 expression in CD4+CD25− T cells, having no significant impact on CD4+CD25+ lymphocytes [25]. The dissimilar impact of both subtypes of breast cancer cells on the expression of immune checkpoints in the co-cultured T cells requires extensive studies in this field to establish relevant immune checkpoint inhibitor-based approaches, which will be highly effective in both mono- and polytherapy of different subtypes of breast cancer.

## 2. Materials and Methods

### 2.1. Preparation of MDA-MB-231 and MCF-7 Breast Cancer Cell Line Cells

MDA-MB-231 (triple-negative) and MCF-7 (ER-positive) breast cancer cell line cells (both adenocarcinomas obtained from the metastatic site; ATCC, Manassas, VA, USA) were thawed from liquid nitrogen, washed twice, and suspended in DMEM medium enriched with FBS and antibiotics. Cells were maintained in culture, 37 °C 5% CO_2_, for the next 4–5 passages to obtain a sufficient number of completely functional cells, thus, cells only from the 4th and 5th passages were used in the study. Prior to the experiment MDA-MB-231 and MCF-7 cells were stained with carboxyfluorescein succinimidyl ester (CFSE, Sigma-Aldrich, Saint Louis, MO, USA) for evaluation of proliferating cells following co-culture. Proliferating cells were analyzed within viable 7AAD- (7-aminoactinomycin D) cells (Appendix A).

### 2.2. Isolation of Peripheral Blood Mononuclear Cells (PBMC) from Healthy Blood Donors

Peripheral blood was obtained by venipuncture from 9 healthy blood donors and subjected to gradient density centrifugation (Histopaque-1077, Sigma-Aldrich) to isolate peripheral blood mononuclear cells (PBMC) (the age of donors was within the range of 26 –35 years; with equal distribution in the context of sex). The obtained PBMC were washed twice in phosphate-buffered saline without calcium and magnesium (PBS, Biomed Lublin) and frozen in cryoprotectant—10% DMSO (DMSO, Sigma-Aldrich) in fetal bovine serum (FBS, Gibco), and stored in liquid nitrogen for the experiments. Prior to co-culture with MDA-MB-231 or MCF-7 cancer cell line cells PBMC were thawed, washed twice, counted, and suspended in DMEM medium enriched with FBS and antibiotic (Gentamicin, Sigma-Aldrich). Due to the use of short co-cultures (up to 24 h) with initially activated PBMC (anti-CD3ε, protocol described in the following subsection), the role of HLA compatibility was excluded in the observed phenomenon, as such mechanisms require longer time intervals, even weeks [26], to develop reactivity in case of lack of histocompatibility. A comparable experimental model is used commonly worldwide and is justified by numerous studies [27,28]. All research was performed in accordance with relevant guidelines/regulations. Informed consent was obtained from all healthy blood donors from whom peripheral blood was collected. The collection of blood was approved by the Local Bioethical Committee at Medical University of Bialystok, approval number R-I-002/61/2015.

### 2.3. Co-Cultures of MDA-MB-231 and MCF-7 Breast Cancer Cell Line Cells with PBMC of Healthy Blood Donors

CFSE-stained MDA-MB-231 or MCF-7 were applied into culture plates (2 × 10^5^ cells/well) and subjected to initial culture to achieve attachment of cells to the surface. Subsequently, PBMC were stimulated with 1 µL/mL of human CD3 epsilon antibody (clone UCHT1) (anti-hCD3ε, R&D Systems, Minneapolis, MN, USA), placed onto MDA-MB-231 or MCF-7 breast cancer cells (2 × 10^6^ of PBMC per well) and incubated for the next 24-h, 37 °C 5% CO_2_. Noteworthy, for each repetition with MDA-MB-231 or MCF-7 cell line, separate PBMC material was used from a single donor—there was no pooling of the PBMC prior application on the wells. Additionally, the experimental layout included MDA-MB-231 of MCF-7 co-cultured with PBMC in the presence of 3 µg/mL anti-CTLA-4 (clone AS32; monoclonal IgG1; company provided reactivity to the extracellular domain of the human CTLA-4) (anti-CTLA-4/-CD152, Invitrogen, Waltham, MA, USA) or 3 µg/mL anti-PD-1 (polyclonal IgG; company provided reactivity to the extracellular domain of the human PD-1) (anti-PD-1/-CD279, R&D Systems) antibodies, and wells with cancer cells alone, cancer cells with 25 ng/mL of colcemid (negative control for proliferation) (Colcemid, Biowest, Nuaille, France) and PBMC alone. The ratio of cancer cells to PBMC and culture time was carefully selected in preliminary experiments, thus obtaining inhibition/killing rate allowing us to monitor additional effects caused by the application of anti-CTLA-4/PD-1. In addition, in the course of the preliminary investigation, we also validated the experimental model of lymphocyte blocking with anti-CTLA-4 or anti-PD-1. In accordance with available literature mentioned in the ‘Introduction’ section, implementation of the CTLA-4 or PD-1 blockage was intended to prevent their inhibitory effects on lymphocytes activation, proliferation, and effector activity (Appendix A—self-made scheme based on information provided in references [29,30,31]. Noteworthy, we confirmed that selected blocking antibodies exerted expected effects on the action of the lymphocytes, demonstrated as facilitated and increased activation (CD69 early-activation marker) and effective production of inflammatory cytokine (IFN-gamma) where immunosuppressive pathway related to CTLA-4 or PD-1 was restrained (Appendix A).

Following incubation, supernatants containing non-adherent cells were aspirated and centrifuged to obtain cells and supernatants for the next experiments. MDA-MB-231 and MCF-7 breast cancer cells were washed twice with PBS and trypsinized with 0.25% trypsin (2.5% Trypsin, Sigma-Aldrich). Detached cells were suspended in an enriched DMEM medium to inactivate trypsin, washed twice in PBS, and used for flow cytometric analysis.

### 2.4. Flow Cytometric Analysis of MDA-MB-231 and MCF-7 Breast Cancer Cell Line Cells Following Co-Cultures

Analysis of MDA-MB-231 cells proliferation status was conducted with the use of CFSE staining and additional staining with 7AAD to exclude dead cells. Negative control of colcemid-treated MDA-MB-231 cells was used to set the gate and evaluate the frequency of proliferating cells. As the CFSE binds covalently to the cytoplasmic proteins and its amount is constant from the beginning, it is evenly distributed in the daughter cells in the course of proliferation. The use of 7AAD dye allowed for additional analysis of viable cell events within studied cancer models. Cell cycle analysis was performed with the use of NSS reagent containing 50 µg/mL propidium iodide (Propidium iodide, PI, Sigma-Aldrich) and 0.035% nonylphenylpolyethylene (Neonidet P40 Substitute, Sigma-Aldrich), followed by the addition of 10 µg/mL RNase solution (RNase A solution, Sigma-Aldrich). Flow cytometric analysis allowed three cell cycle phases to be distinguished: G1-phase, S-phase, and G2-phase. Proliferation and cell cycle data were acquired with the use of a FACS Calibur flow cytometer (BD Bioscience, Franklin Lakes, NJ, USA) and subsequent analysis was performed using FlowJo software (TreeStar Inc., Ashland, OR, USA). Combination of viability, proliferation, and cell cycle assessment allowed for the complex determination of the function-related events within breast cancer cells in the described co-culture conditions (Appendix A).

### 2.5. Flow Cytometric Analysis of PBMC after Co-Culture with MDA-MB-231 and MCF-7

Lymphocytes present in supernatants of PBMC and MDA-MB-231 or MCF-7 cells co-culture were stained with fluorochrome-conjugated monoclonal antibodies including: anti-CD3 FITC (clone SK7), anti-CD3 PerCP (clone SK7), anti-CD4 PerCP (clone SK3), anti-CD8 FITC (clone HIT8a), anti-CD152/CTLA-4 APC (clone BNI3), anti-CD279/PD-1 PE (clone MIH4) (BD Bioscience). Moreover, immune checkpoint proteins were studied, both on the surface of lymphocytes and MDA-MB-231/MCF-7 with the use of anti-CTLA-4 APC, anti-PD-1 PE, anti-PD-L1 PE (clone MIH1), anti-CD80 BB515 (clone L307.4), and anti-CD86 PE-Cy5 (clone IT2.2). In addition, following co-culture with MDA-MB-231 or MCF-7, cancer cells supernatants containing non-adherent lymphocytes were subjected to additional 6h stimulation with PMA, ionomycin, and Brefeldin A (Leukocyte Activation Cocktail, BD Bioscience) to evaluate the effects of the presence of breast cancer cells on the production of IFN-gamma and IL-10 by lymphocytes. Cells were stained with the following monoclonal antibodies for staining extracellular markers: anti-CD4 PerCP (clone SK3), anti-CD8 APC (clone SK1), anti-CD3 FITC (clone SK7), and anti-IFN-gamma PE (clone 4S.B3) or anti-IL-10 PE (clone JES3-19F1) after the permeabilization step to assess intracellular cytokines. Intracellular staining procedures were also implemented following co-culture with cancer cells to evaluate perforin (AlexaFluo647-conjugated, clone δG9) and granzyme B (PE-conjugated, clone GB11) production by lymphocytes. Flow cytometric data were acquired with the use of a FACS Calibur flow cytometer (BD Bioscience, USA) and subsequent analysis was performed using FlowJo software (TreeStar Inc., Ashland, USA). Production of IFN-gamma and IL-10 to evaluate inflammatory cytokines crucial in the context of effector function regulation, and perforin and granzyme B to establish lymphocyte-related cytotoxic function were assessed within CD4+, CD8+ and the total pool of lymphocytes (with anti-CD3 PE-Cy7 and anti-CD8 FITC antibodies implemented) (Appendix A). Evaluation of CTLA-4 and PD-1 levels within CD4+ and CD8+ lymphocyte subsets included frequencies of CTLA-4- and PD-1-expressing cells and mean fluorescence intensities (MFI) within these populations (Appendix A). An in-depth description of the antibodies used, their specificity, and representative dot plots and histograms including FMO controls were presented within Appendix A. Thus, effective binding to the studied targets and their specific reactivity was confirmed (Appendix A).

### 2.6. Colorimetric Assessment of LDH Activity-Based Cytotoxicity

Supernatants from co-cultures of MDA-MB-231 or MCF-7 cancer cells with PBMC, and inhibition of CTLA-4 or PD-1, were subjected to analysis of LDH activity following 24-h incubation to evaluate the cytotoxicity of the described layouts in reference to breast cancer cells. Analysis was performed with the use of the Pierce LDH Cytotoxicity Assay Kit (Thermo Scientific, Rockford, IL, USA). In accordance with the kit instruction, following 30-min incubation with Reaction Mixture, Stop Solution was added and absorbance was measured at 480 and 680 using the Epoch plate-reader (BioTek Instruments, Winooski, VT, USA) to determine LDH activity. Data were presented as LDH activity-associated absorbance after subtraction of values measured in blank control—enriched cell culture medium used in the experiments. An increase in LDH activity in the tested supernatants indicated its release from cancer cells (as there was no change in the viability of PBMC in flow cytometric data—data not shown), and which presence extracellularly is always associated with disturbances in the membrane-cell damage.

### 2.7. Statistical Analysis

Due to the non-normal distribution of the data, the non-parametric Wilcoxon signed-rank test was used for statistical analysis of the paired results, and Mann-Whitney for unpaired data. Estimation of the Gaussian distribution of the data was performed with the use of Shapiro–Wilk and D’Agostino Pearson normality tests. Two independent tests were required to confirm the normality of the distribution, otherwise, data were further tested with non-parametric tests. Acquired flow cytometric and colorimetric data were presented on graphs as a relative percentage compared to MDA-MB-231/MCF-7 or PBMC alone defined as initial 100%. Statistics were generated using GraphPad Prism software (GraphPad Software Inc., San Diego, CA, USA). Statistical differences were indicated with asterisks: * *p* < 0.05, ** *p* < 0.01, *** *p* < 0.001, **** *p* < 0.0001, or values for non-significant data but showing tendency and vales close to *p*-value of 0.05. As described within the figures, statistically significant differences were indicated when the selected group was compared to cells alone (black values/asterisks) or cells co-cultured with PBMC (light grey values/asterisks). Additional brackets above two cell types within certain groups were applied to clearly demonstrate the presence of significant differences in response between MCF-7 and MDA-MB-231 cell lines if present.

## 3. Results

### 3.1. Effects of CTLA-4 and PD-1 Blockage on Proliferation Status of MDA-MB-231 and MCF-7 Cells in Presence of PBMC

Despite the lack of changes in the proliferation rate of MDA-MB-231 cancer cells in the presence of anti-CD3-stimulated PBMC alone following 24-h co-culture, we found that inhibition of CTLA-4 on the surface of mononuclear cells allowed for significant reduction of proliferating cancer cells, even when compared to MDA-MB-231 affected by lymphocytes alone. No changes in the proliferation status of breast cancer cells were observed with reference to additional inhibition of PD-1 in co-culture with PBMC. Interestingly, the response of MCF-7 cancer cells to CTLA-4 blockage was significantly different compared to the MDA-MB-231 cell line. In addition, no changes were observed in other studied layouts, both when compared to MCF-7 alone or in co-culture with PBMC (Figure 1a). Assessment of viable cell events related to MCF-7 or MDA-MB-231 cancer cells revealed that only the latter one responded with lower viable cells number compared to co-culture with PBMC when blockage of PD-1 was introduced (Figure 1d).

### 3.2. Cell Cycle Status of MDA-MB-231 or MCF-7 Cells in Presence of PBMC Alone, and Additional Inhibition of CTLA-4 and PD-1

In accordance with the results observed in the proliferation assay, inhibition of CTLA-4 on the surface of PBMC allowed for the significant arrest of MDA-MB-231 cancer cells in the G1/S-phase of the cell cycle. In addition, despite the lack of significant changes in proliferation status, the presence of PBMC alone and in combination with PD-1-blockage allowed for the significant arrest of breast cancer cells in the G1/S-phase. In accordance with proliferation data, no crucial changes were demonstrated in the context of MCF-7 breast cancer cells with only a slight increase in G1-phase cells when anti-PD-1 was applied (Figure 1b).

### 3.3. Cytotoxicity of PBMC Alone, and Additional Inhibition of CTLA-4 and PD-1 with Reference to MDA-MB-231 or MCF-7 Cells

Assessment of LDH activity in supernatants of cultured MDA-MB-231 or MCF-7 breast cancer cells revealed significant cytotoxic activity of PBMC on these cells. Importantly, more pronounced effects were observed in the additional application of CTLA-4 inhibition, with approximately 32% higher LDH activity in supernatants compared to MDA-MB-231 treated with PBMC only. We did not find significant differences between MDA-MB-231 cells treated with PBMC alone versus treatment with PBMC and the additional inhibition of PD-1. Furthermore, despite the effective activity of PBMC against MCF cancer cells, it was less pronounced compared to that observed in MDA-MB-231 cells. Moreover, even though cytotoxicity was reported in the MCF-7 cell line in the settings tested, none of the inhibitory antibodies applied-anti-CTLA-4 or anti-PD-1, caused significant effects on the viability of MCF-7 cells co-cultured with PBMC only (Figure 1c).

### 3.4. Differential Expression of Immune Checkpoint Proteins on the Surface of Tested PBMC and MDA-MB-231/MCF-7 Breast Cancer Cells

Analysis of selected immune checkpoint proteins on the breast cancer cells surface revealed significant differences in the pattern of expressed markers. In reference to MDA-MB-231 cells, PD-L1 expression was found to dominate above all in the context of both the frequency of cells and mean expression on the surface. The frequency of PD-1 was also found to be slightly increased, however, its levels were found to stand out predominantly in MCF-7 cancer cells. A certain percentage of MCF-7 cells was also reported to express CD86, nonetheless, it was PD-1 that dominated in these cells. Noteworthy, PD-1 and PD-L1 levels predominantly were significantly different between MDA-MB-231 and MCF-7 breast cancer cells (Figure 2a). 

In the context of immune checkpoint proteins levels within lymphocytes, we found that the PD-1 marker is dominant above all analyzed, with a surprisingly small contribution of CTLA-4 in the context of frequency in studied cell populations. Noteworthy, CD8+ lymphocytes demonstrated significantly higher frequencies when compared to CD4+ T cells or even the total pool of lymphocytes. Interestingly, despite the lack of distinction in the context of frequency, CD80 marker mean expression was different between CD8+ and CD4+ or total lymphocytes, thus, providing a potential target for other further studies (Figure 2b).

### 3.5. IFN-Gamma and IL-10 Production by Lymphocytes in Response to MDA-MB-231/MCF-7 Presence, and with Additional Inhibition of CTLA-4 or PD-1

First, it is worth noting that significant effects of cancer cells’ presence and applied treatment were only reported in reference to co-cultures with MDA-MB-231 breast cancer cells. No changes were demonstrated in cell cultures involving MCF-7 use, with only a slight tendency for higher levels of IL-10-producing CD8+ lymphocytes in co-culture with PBMC and anti-PD-1 presence. In addition, changes in IFN-gamma and IL-10 production reported in MDA-MB-231 were significantly different compared to MCF-7, predominantly through influence on CD8+ T cell population in the context of IL-10 and CD4+ lymphocytes as far as IFN-gamma was concerned. We found that inhibition of CTLA-4 or PD-1 on the surface of PBMC in co-culture with MDA-MB-231 significantly reduced the ability of lymphocytes to produce IFN-gamma in response to additional stimulation with PMA and ionomycin. The decline in IFN-gamma production seems to be associated predominantly with reduced production of that cytokine within CD4+, but not to the same extent in CD8+ lymphocytes. Changes in IFN-gamma apply not only to intracellular production but are also reflected in the decreased number of IFN-gamma-producing CD4+ lymphocytes in the presence of MDA-MB-231 alone, or with additional CTLA-4 or PD-1 inhibition. In addition, despite a lack of changes in intracellular production, the decline was also observed in the frequency of IFN-producing CD8+ lymphocytes when CTLA-4 was blocked. Regarding IL-10 production within lymphocytes, its reduced values were found in lymphocytes cultured in the presence of MDA-MB-231 alone, and additional CTLA-4 or PD-1 inhibition. Interestingly, such decline was shown to be associated with decreased production of IL-10 within CD8+, but not in CD4+ lymphocytes. In addition, effects exerted by PD-1 blockage seem to be more pronounced compared to co-culture with MDA-MB-231 alone or in combination with CTLA-4 inhibition. Furthermore, a significant decline in IL-10 production by CD8+ lymphocytes was accompanied by lower frequencies of IL-10-producing CD8+ lymphocytes (Figure 3a–c).

### 3.6. Alterations in CTLA-4 and PD-1 on the Surface of Lymphocytes in Co-Culture with MDA-MB-231/MCF-7 and Additional Presence of CTLA-4 or PD-1 Inhibition

We found that inhibition of PD-1 in co-culture with MDA-MB-231 led to a decrease in the frequency of PD-1+ cells within CD4+ and CD8+ lymphocyte populations, as well as a decrease in PD-1 expression in both subsets. Furthermore, a decline in the frequency of CD8+PD-1+ lymphocytes was observed in the presence of CTLA-4 or PD-1 blockage. Additionally, inhibition of CTLA-4 led to a significant decrease in CD4+CTLA-4+ lymphocytes. Interestingly, contrary to MDA-MB-231, the presence of MCF-7 was mostly associated with completely different effects exerted on lymphocyte’s expression of CTLA-4 or PD-1. We found, inter alia, that MCF-7 cells caused an increase in CTLA-4 levels within CD4+ and CD8+ lymphocytes. Noteworthy, implementation of anti-PD-1 seemed to diminish that effect in the context of PD-1 expression on CD4+ and CD8+ lymphocytes. The most pronounced influence of MCF-7 breast cancer cells on lymphocytes was found when analyzing CD4+CTLA-4+/PD-1+ and CD8+CTLA-4+/PD-1+ frequencies in total lymphocytes. In accordance with general PD-1 expression, here, PD-1 blockage also reduced the fraction of PD-1-expressing CD4+/CD8+ lymphocytes. Additionally, a slight decrease was observed in the population of CD4+ and CD8+ with CTLA-4 when the inhibition of CTLA-4 was applied (Figure 4a–c).

### 3.7. Changes in Perforin and Granzyme B Production by Lymphocytes in the Course of Co-Culture with MDA-MB-231/MCF-7 and Implementation of CTLA-4 and PD-1 Inhibition

Evaluation of perforin and granzyme B within the total pool of lymphocytes revealed an increase in both proteins in response to the anti-PD-1 application in the co-culture of MDA-MB-231. In the context of the response of CD4+ lymphocytes to PD-1 blockage, significantly higher levels were observed predominantly in the granzyme B level. Importantly, the highest differences were found in perforin production when PBMC were co-cultured with MDA-MB-231 and anti-PD-1 leading to the higher frequency of CD8+Perforin+lymphocytes. Slight changes were also observed in these populations of cells when anti-CTLA-4 was implemented. In contrary to MDA-MB-231, the presence of MCF-7 did not cause essential changes in lymphocytes in the context of granzyme B or perforin production (Figure 5a–c).

## 4. Discussion

Cytotoxic T lymphocyte-associated antigen 4 (CTLA-4) and programmed cell death protein 1 (PD-1), both representing the immune checkpoint receptor family, are essential factors in maintaining homeostasis of the immune system through negatively regulating immune responses, providing self-tolerance [29]. This phenomenon can be also observed in the tumor microenvironment, where the overexpression of inhibitory immune checkpoints causes the depletion in anti-tumor activity resulting in tumor cells’ proliferation and metastasis [30]. Therefore, inhibiting the axis between cancer and immune cells through antibodies directed at CTLA-4/PD-1 leads to the invaluable improvement in treatment efficacy of numerous malignant tumors especially melanoma, lung cancer, and breast cancer [6,31]. The effectiveness of CTLA-4 blocking is supported by studies demonstrating prolonged cells activation and sustained inflammation [32]. Here we implemented in vitro model involving breast cancer cell lines (MDA-MB-231 and MCF-7) and healthy PBMC (Appendix A; model justified by numerous studies [27,28] to reveal how blocking of CTLA-4 or PD-1 might contribute to anti-cancer effects, and whether its support initially presumed hypothesis. 

In accordance with previous results suggesting the favorable role of anti-CTLA-4 antibodies in supporting gemcitabine-based suppression of tumor growth in mice (with anti-CTLA-4 alone exhibiting a slight hampering effect) [17], we found that CTLA-4 blockage effectively supports the response of immune cells against cancer cells. However, here we demonstrated that not only type of breast cancer is essential to obtain the same response but also individual features of cancer cells. The triple-negative model of breast cancer was the one responding with growth inhibition to additional implementation with anti-CTLA-4. Such changes were not found when MCF-7 cells (ER-positive) were used, thus suggesting the need for careful consideration of immunotherapy application in breast cancers. In the study of Peng et al., the molecular analysis of CTLA-4 expression on different subtypes of breast cancer cells showed the highest CTLA-4 level in TNBC, which seems to be a direct confirmation of our results [33].

Analysis of immune checkpoint proteins on tested breast cancer cells revealed the difference between them also in the context of these molecules apart from hormone receptors. The obtained data demonstrated the dominance of PD-L1 in MDA-MB-231 and PD-1 in breast cancer cells. In addition, PD-1 seemed to be the predominant checkpoint protein. Despite low levels of CTLA-4 on the surface of the lymphocytes, its inhibition was still more effective in reducing MDA-MB-231 cell proliferation than PD-1 blockage. These data suggest the need for future mechanistic research to comprehensively study mutual interactions between immune checkpoints and breast cancer cells modulation in the immunotherapeutic application.

In experiments with dendritic cells transfected with anti-CTLA-4 antibody mRNA, the activity of this immune modulator was associated with the enhancement of antigen-specific cytotoxic T lymphocytes (CTL) responses against breast cancer cells, including MCF-7, MDA-MB-231, and T4D7 cell lines [34]. On the contrary, we observed that application of anti-CTLA-4 (and PD-1 partially) antibodies into the tumor micro-environment inhibits CD4+ and CD8+ cell activity affecting their interaction with breast cancer cells. Here, however, changes were only observed in MDA-MB-231 and not MCF-7 breast cancer cells. In the context of PD-1 inhibition, lack of changes in MDA-MB-231 breast cancer cell proliferation would appear to be in accordance with previous data indicating no effects on survival and metastasis of tumors treated only with anti-PD-1 antibodies [20]. Interestingly, blockage of PD-1 was still found to affect cancer cells viability compared to the co-culture with PBMC alone. Additionally, MCF-7, as in the case of proliferation, did not demonstrate an unambiguous response to selected immunotherapeutic in the context of viability. 

The results of our study on the effects of breast cancer cells on the cytokine production profile of CD4+ and CD8+ lymphocytes are reflected in research conducted recently in animal models by da Cunha et al. Both research teams demonstrated that the presence of breast cancer cells did not affect the frequency of IL-10-producing CD4+ T cells, but instead significantly reduced IL-10-positive CD8+ T cells. Furthermore, in both CD4+ and CD8+ T cells, interaction with tumor cells did not influence the production of IFN-gamma [35]. Interestingly, we found that ER-positive MCF-7 cancer cells unlike MDA-MB-231 did not respond with a decline in IFN-gamma or IL-10 production in lymphocytes. Furthermore, previously obtained data from other research teams has indicated that failure of NAC treatment and lack of complete response was significantly associated with the immunosuppressive activity of IL-10 [36] Interestingly, we found that the application of CTLA-4 blocking antibodies effectively reduced the production of IL-10 within lymphocytes, predominantly through a decline in IL-10+ CD8+ cell frequency. Therefore, blockage of CTLA-4 on lymphocytes might allow for overriding adverse suppression of the anti-tumor immune response through reduction of cells bearing suppressive potential, including IL-10-producing Th2 cells, or possibly even Foxp3+ regulatory T cells or IL-10-producing regulatory B cells. Such a favorable role of immunosuppressive Foxp3+ T cell depletion was demonstrated previously in a mouse model of mammary carcinoma where reduced immunosuppressive activity significantly affected tumor progression through increased cancer cell death and improved overall survival [37]. In line with this, Pruitt et al. showed that the presence of immune checkpoint modulators inhibits induction of regulatory T cells and related immunosuppressive effects [34]. In addition to changes in IFN-gamma or IL-10, granzyme B/perforin-related cytotoxic effects were clearly detected when anti-PD-1 (partially anti-CTLA-4) was implemented in co-culture of PBMC with MDA-MB-231. Importantly, no changes were found when the ER-positive MCF-7 model was tested. In general, however, features of breast cancer must be carefully monitored before any therapeutic approach will be implemented as these might affect the effectiveness of the response to the immunotherapy. 

Experiments conducted by Chen et al. showed that inhibition of anti-tumor response might not only occur through the interaction of tumor receptors with T cell-expressed CTLA-4, but also via suppression of dendritic cell function as a consequence of their interaction with CTLA-4+ breast cancer cells. Furthermore, blockage of CTLA-4 on breast cancer cells led to the restoration of dendritic and T cells activity with concomitant inhibition of cancer cell proliferation [38]. These data are in accordance with the results obtained in our study where CTLA-4 inhibition led to a significant reduction of breast cancer cell proliferation. However, we consider the blocking of anti-tumor activity is associated with cancer cell interaction with CTLA-4+ lymphocytes, thus, making the T cells of crucial importance, as far as MDA-MB-231 cells are considered. In the study by Kaewkangsadan et al., it has been demonstrated that unlike intratumoral infiltration by CD4+ and CD8+ T cells, tumor-infiltrating CTLA-4+ and PD-1+ T cell levels were not associated with pathological complete response (pCR) following neoadjuvant chemotherapy (NAC) in large and locally advanced breast cancers (LLABC) [36]. In the present study, we have shown that blockage of CTLA-4 receptor on lymphocytes and concomitant reduction of CD4+CTLA-4+ and CD8+PD-1+ cells seem to be a crucial element in inhibition of MDA-MB-231 breast cancer cell proliferation and their arrest in the G1-phase. 

Additionally, our results, in combination with data from Page et al. indicate that application of anti-CTLA-4 antibodies is likely not only to increase the number of infiltrating T cells in the tumor environment showed by their team, but also to reduce the proportion of unfavorable immunosuppressive populations of CTLA-4+ lymphocytes as demonstrated by change of lymphocytes phenotype in our study [11]. In contrast, PD-1 blocking did not affect the proliferation status of MDA-MB-231 cells, even though it effectively reduced the frequency of PD-1+ CD4+ and CD8+ lymphocytes. It is worth noting that the reduced frequency of PD-1-expressing lymphocytes might have favorable effects on avoiding tumor-induced PD-L1-driven suppression of the immune response which has been previously suggested to be responsible for the reduced survival of breast cancer patients [18]. Other research teams have indicated the possible association between the number of PD-1-positive T lymphocytes infiltrating the tumor environment with worse overall survival rate in breast cancer patients [39,40]. With reference to PD-1 expression within lymphocyte subsets, slight differences in response to anti-PD-1 blockage seem to be associated with differential expression of PD-1 in tumor-associated lymphocytes, predominantly in CD4+ T cells [41]. In fact, PD-1 expression is different between CD4+ and CD8+ lymphocytes but here we demonstrated that the latter population exhibit the highest levels of PD-1, with relatively lower levels of CTLA-4 in both cell types simultaneously. Noteworthy, the presence of MCF-7 cells led to completely different changes compared to MDA-MB-231 in reference to CTLA-4 or PD-1 expression. Co-culture of PBMC with MCF-7 breast cancer cells is causing an increase in CTLA-4 and PD-1 expression within lymphocytes. In the study of Saleh et al., the MCF-7 breast cancer cells greatly upregulated the PD-1 expression in CD4+CD25− T cells, having no significant impact on CD4+CD25+ lymphocytes [25]. Furthermore, we reported that anti-PD-1 application predominantly diminished these effects. Cumulatively, such an increase in selected immune checkpoint proteins might be a possible explanation of the different responses of MCF-7 cancer cells when compared to MDA-MB-231.

Unlike Sagiv-Barfi et al. who demonstrated the effects of a lack of PD-L1-blockage IFN-gamma production in CD44+ central memory T cells(19), here we have shown a significant reduction of IFN-gamma-producing T cells (exclusively CD4+ lymphocytes) in response to anti-PD-1 and anti-CTLA-4 inhibition. Such phenomenon, however, was only observed in triple-negative MDA-MB-231 cancer cells, not MCF-7. Another research group showed that the supportive potential of CTLA-4 blockage in HER2/Neu antibody and triciribine combination treatment of mouse mammary tumors was associated with increased concentrations of IFN-gamma in tumor lysates and plasma [42]. Interestingly, Sagiv-Barfi et al. reported that PD-L1 inhibition combined with ibrutinib treatment (in A20 lymphoma cell line) led to increased frequency of IFN-gamma-producing CD44+ central memory T cells [19]. Likewise, the application of anti-PD-1 with anti-GITR antibodies synergistically induced the production of IFN-gamma in tumor-associated lymphocytes. However, in accordance with our data, PD-1 monotherapy did not affect the frequencies of IFN-gamma-producing CD8+ T cells [43]. Moreover, we found that cytotoxicity against MDA-MB-231 breast cancer cells was associated with an increase in granzyme B and perforin B. As indicated above, such effect was not obtained in the MCF-7 cancer cell line co-cultured with PBMC. To summarize, we assume that the observed reduction in the number of IFN-gamma-producing CD4+ T cells might be simultaneously associated with an increased release of IFN-gamma into the cancer cell micro-environment in triple-negative MDA-MB-231 cancer cells. It was followed additionally by an increase in cytotoxicity-related perforin/granzyme B within lymphocytes. It is worth noting that previous studies in mouse models of breast cancer demonstrated that repetitive administration of PD-1/PD-L1 axis-blocking antibodies can cause fatal hypersensitivity reactions [44]. Based on the obtained results, we presume that such phenomenon might be associated with IFN-gamma reduction within lymphocytes following anti-PD-1 administration, which in fact has been shown in numerous studies to play a protective role in the context of excessive immune responses [45,46]. Considering observed differences in immune checkpoint proteins in tested breast cancer cell lines, appropriate selection of these molecules’ inhibitors might decrease these adverse effects, maintaining anti-cancer effects adjusted to cancer type.

## 5. Conclusions

The obtained results indicate the advantageous properties of CTLA-4 blockage within lymphocytes in the suppression of breast cancer cell proliferation. However, our data revealed that responses of cancer cells might be different depending on surface expression of hormone receptors, or as indicated in here, variations in immune checkpoint molecules. Thus, further experiments would be crucial to carefully consider the therapeutic anti-tumor potential of CTLA-4 modulation in the context of breast cancer treatment. In addition, combining immune checkpoint inhibitors with factors amplifying their effects is another issue of current investigation. One of them is demonstrated in an animal model blockage of CD73 leading to improved efficiency of anti-CTLA-4- or anti-PD1-induced antitumor activity in colon, prostate, and breast cancer subjects [47]. Here demonstrated the contribution of other checkpoint molecules such as CD80 suggest that studies on the combinatory application of these protein inhibitors might be of great importance, especially considering different features of only seemingly the same types of breast cancers. Furthermore, recent data has been shown to be promising regarding anti-SEMA4D antibody (semaphorin 4D, CD100) and its enhancing properties in relation to immunomodulation of the anticancer ability of anti-CTLA4/PD-1 antibody activity [48]. This data has been crucial in shedding new light on novel therapeutic approaches involving a combination of CD100 and immune checkpoint blocking antibodies.

## Figures and Tables

**Figure 1 cells-10-02044-f001:**
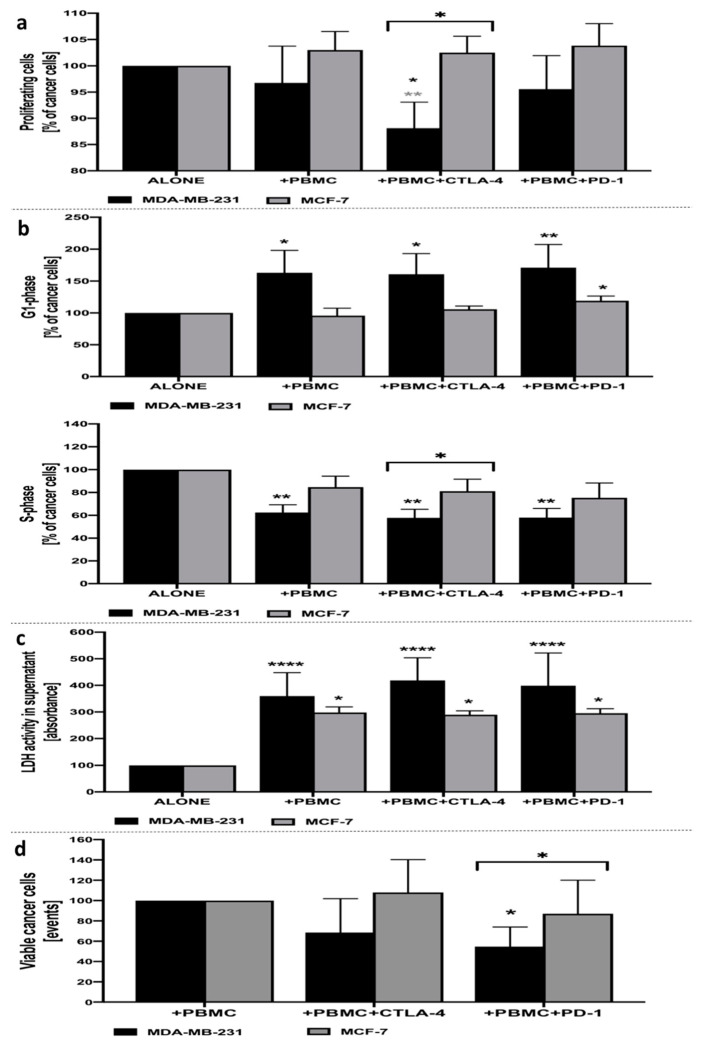
Analysis of MDA-MB-231 and MCF-7 breast cancer-associated parameters. Proliferation status of CFSE-labeled MDA-MB-231/MCF-7 breast cancer cells following 24-h incubation with PBMC alone, and with the addition of CTLA-4 or PD-1 inhibitors (**a**). Cell cycle status of MDA-MB-231/MCF-7 breast cancer cells following 24-h incubation with PBMC alone, and with the addition of CTLA-4 or PD-1 inhibitors, based on the frequency of cancer cells in G1- (upper graph) and S-phase (lower graph) (**b**). Cytotoxic effects of PBMC alone, and with the addition of CTLA-4 or PD-1 inhibitors, on MDA-MB-231/MCF-7 breast cancer cells following 24-h incubation, based on the activity of LDH in supernatants (**c**) (*n* = 10). Evaluation of viable cancer cells based on dead cells exclusion with 7AAD (**d**). Asterisks above individual cancer cell line columns show the level of significance with colors indicating whether the difference is reported when the specific column is compared to cancer cells alone (black) or co-cultured with PBMC (light grey) (* *p* < 0.05, ** *p* < 0.01, *** *p* < 0.001, **** *p* < 0.0001). Moreover, brackets with significance demonstrated are applied when there was a difference in response between MDA-MB-231 and MCF-7 cells.

**Figure 2 cells-10-02044-f002:**
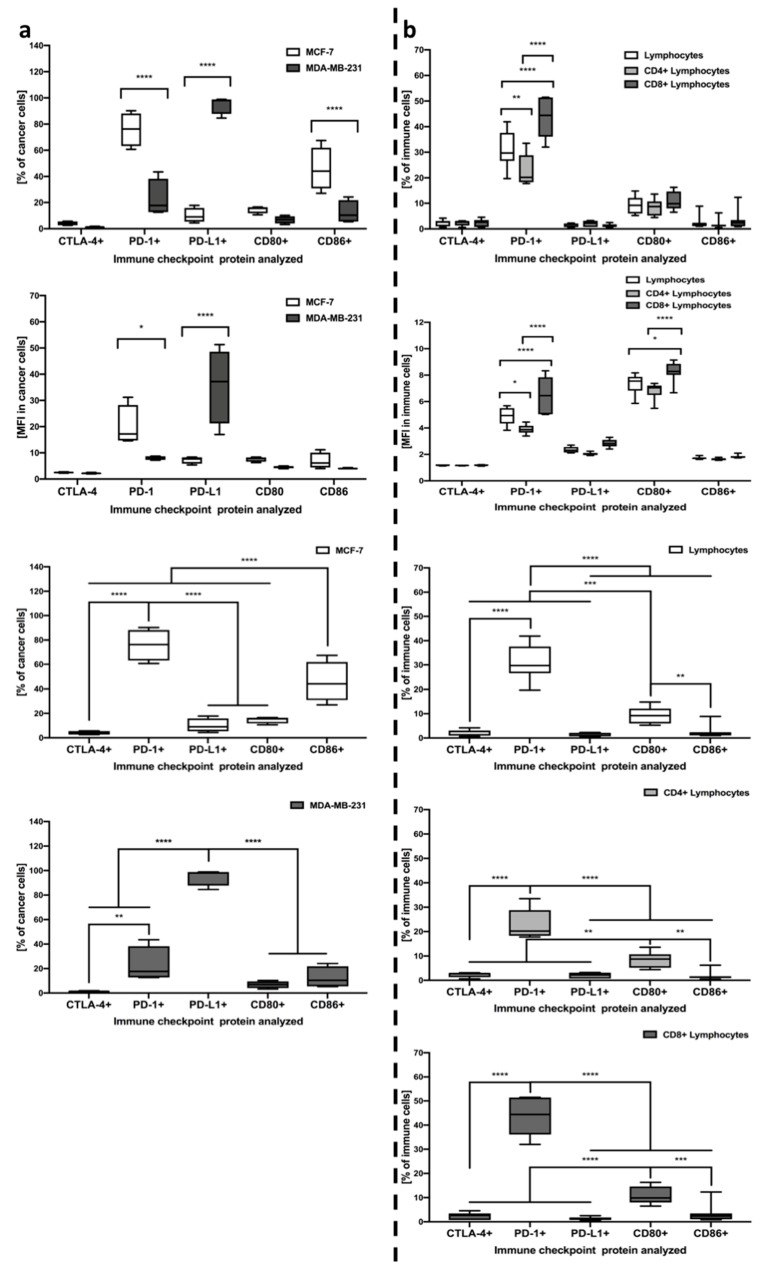
Evaluation of CTLA-4, PD-1, PD-L1, CD80, and CD86 immune checkpoint proteins on the studied cells. Acquired data included frequency of cells expressing selected markers and mean expression (MFI) on single cells, analyzed within MDA-MB-231/MCF-7 cancer cells (*n* = 8) (**a**) and lymphocytes including CD4+, CD8+, and the total pool of lymphocytes (*n* = 7) (**b**). The first two rows demonstrate significant differences between MDA-MB-231/MCF-7 or different lymphocyte subsets within each of the immune checkpoint proteins tested. Left column, rows 3–4, demonstrate the profile of immune checkpoint proteins within MCF-7 and MDA-MB-231, respectively. In the right column, rows 3–5, differences between studied checkpoint proteins are indicated for all studied lymphocyte subsets separately (statistically significant differences are indicated with asterisks) (* *p* < 0.05, ** *p* < 0.01, *** *p* < 0.001, **** *p* < 0.0001).

**Figure 3 cells-10-02044-f003:**
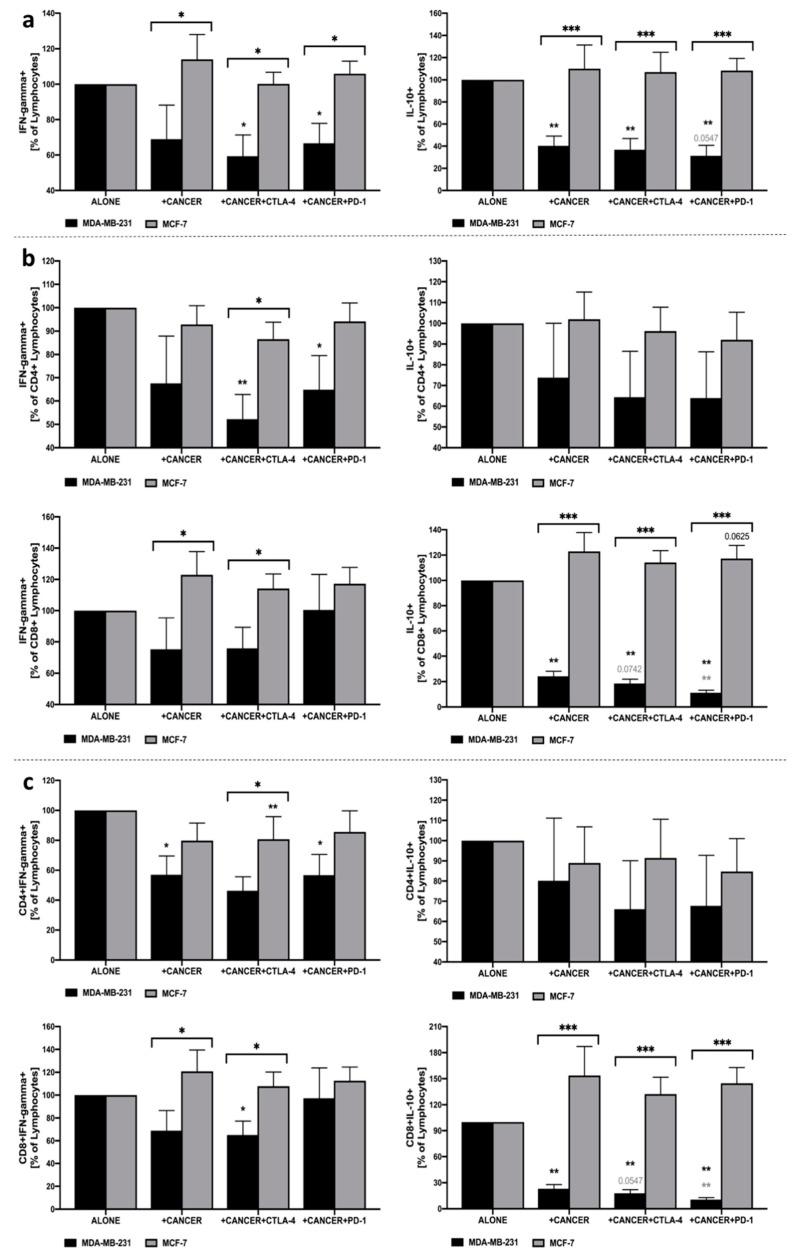
Analysis of IFN-gamma and IL-10 production within lymphocytes co-cultured with MDA-MB-231/MCF-7. Production of IFN-gamma and IL-10 in the total pool of lymphocytes (**a**), within CD4+ and CD8+ lymphocyte subsets (**b**), and frequencies of IFN-gamma- and IL-10-producing CD4+ and CD8+ lymphocytes (**c**) (*n* = 8). Asterisks above individual cancer cell line columns show the level of significance with colors indicating whether a difference is reported when the specific column is compared to cancer cells alone (black) or co-cultured with PBMC (light grey) (* *p* < 0.05, ** *p* < 0.01, *** *p* < 0.001, **** *p* < 0.0001). Moreover, brackets with significance demonstrated are applied when there was a difference in response between MDA-MB-231 and MCF-7 cells.

**Figure 4 cells-10-02044-f004:**
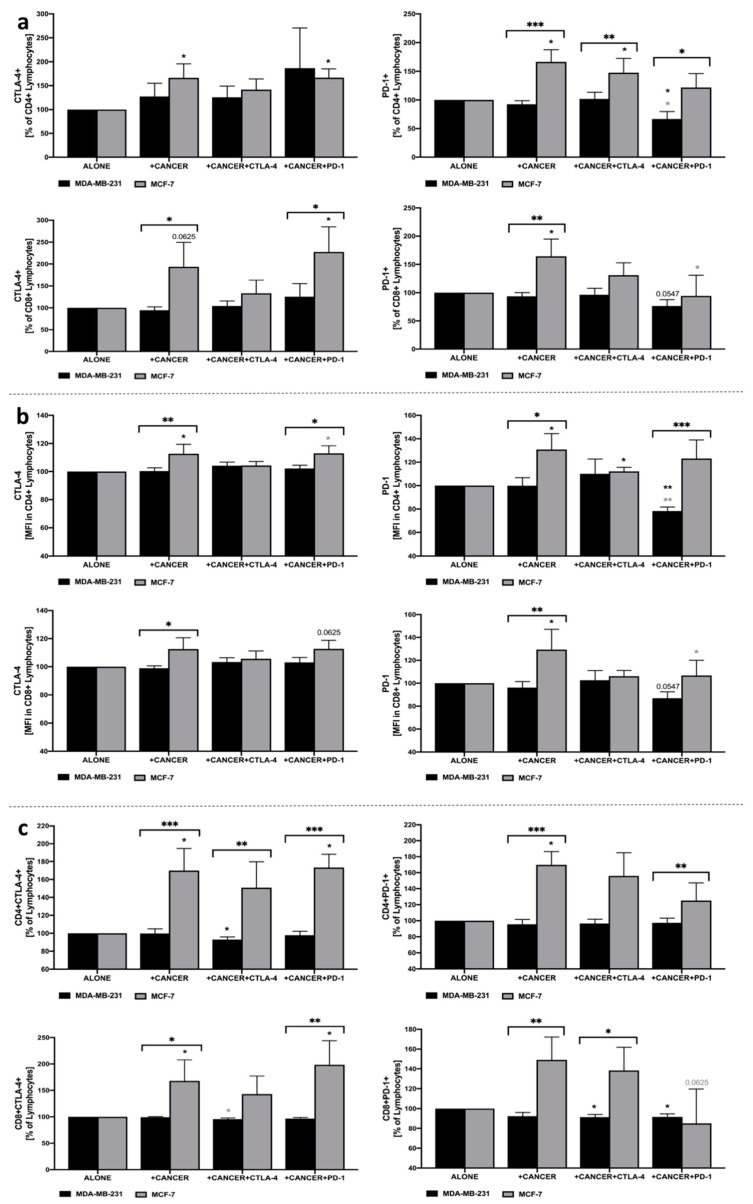
Evaluation of CTLA-4 and PD-1 levels in lymphocytes in response to co-culture with MDA-MB-231/MCF-7. Changes in frequency of CTLA-4+ and PD-1+ cells within CD4+ and CD8+ lymphocytes (**a**), expression of these markers presented as mean fluorescence intensity (MFI) (**b**), and percentage of CTLA-4+ or PD-1+ CD4+ and CD8+ cells within total pool of lymphocytes (**c**) (*n* = 8). Asterisks above individual cancer cell line columns show the level of significance with colors indicating whether a difference is reported when the specific column is compared to cancer cells alone (black) or co-cultured with PBMC (light grey) (* *p* < 0.05, ** *p* < 0.01, *** *p* < 0.001, **** *p* < 0.0001). Moreover, brackets with significance demonstrated are applied when there was a difference in response between MDA-MB-231 and MCF-7 cells.

**Figure 5 cells-10-02044-f005:**
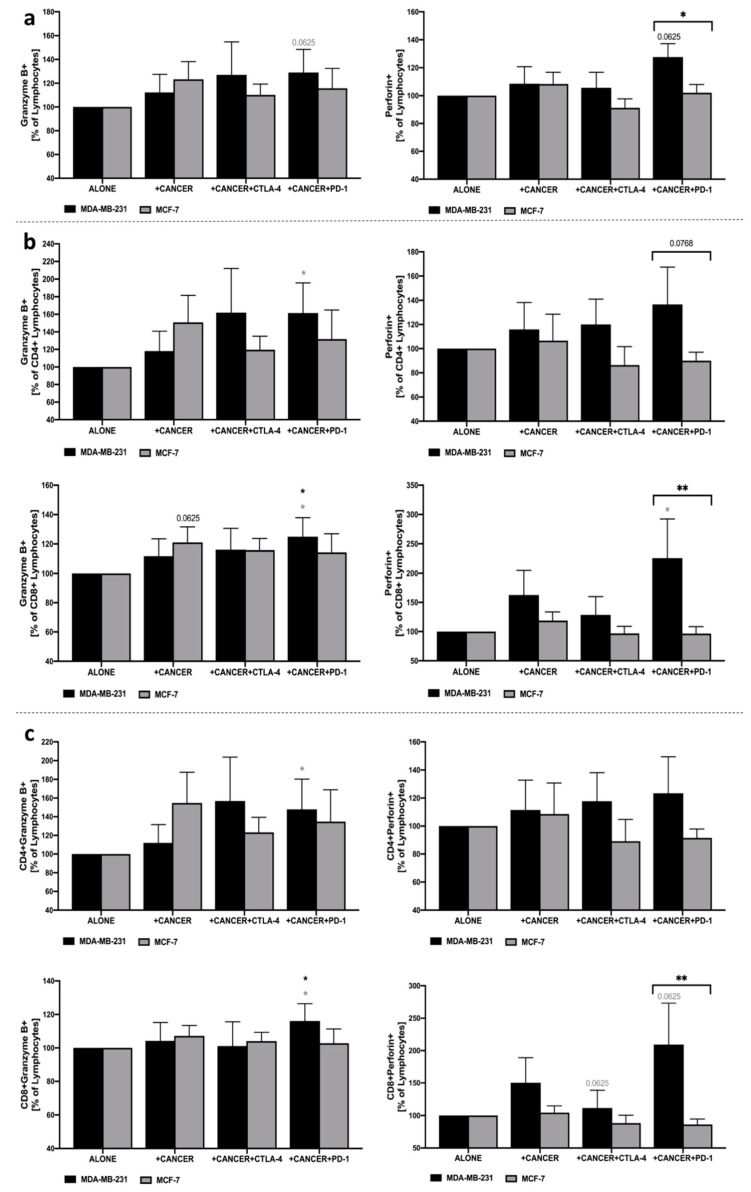
Granzyme B and perforin production in PBMC following co-culture with MDA-MB-231/MCF-7 and CTLA-4/PD-1 blockage. Changes in frequency of granzyme B+ and perforin+ cells were presented as the frequency within total lymphocytes (**a**), CD4+ and CD8+ lymphocytes (**b**), and percentage of granzyme B+ or perforin+ CD4+/CD8+ cells within the total pool of lymphocytes (**c**) (*n* = 6). Asterisks above individual cancer cell line columns show the level of significance with colors indicating whether the difference is reported when the specific column is compared to cancer cells alone (black) or co-cultured with PBMC (light grey) (* *p* < 0.05, ** *p* < 0.01, *** *p* < 0.001, **** *p* < 0.0001). Moreover, brackets with significance demonstrated are applied when there was a difference in response between MDA-MB-231 and MCF-7 cells.

## Data Availability

The data presented in this study are available on request from the corresponding author. The data are not publicly available due to privacy restrictions.

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
