# Peer review of "Differential Response of MDA-MB-231 and MCF-7 Breast Cancer Cells to In Vitro Inhibition with CTLA-4 and PD-1 through Cancer-Immune Cells Modified Interactions"

_cells, 2021, doi:10.3390/cells10082044_

Round 1

Reviewer 1 Report

The authors have tried to evaluate the effect of antiCTLA-4 and PD-1 in two breast cancer cell lines even though their results are interested the article needs careful analysis.

  1. In section 2.1 from the results the authors say in MCF-7 cell line there is no significant change in viability when the cells are treated with PBMC and PD-1 blocker, but in Figure 1d there is a * at this column as if there are statistically significant changes.
  2. In section 2.3 from results the authors say: "Moreover, none of the inhibitory antibodies applied -anti-CTLA4 or anti-PD-1, causes significant effects in reference to MCF-7 cell viability (Fig 1C)" but in Figure 1c there are significant results also for MCF-7 data in cells cocultured with PBMC, PBMC+CTLA-4 and PBMC+PD-1.
  3. All figures need to be better explained, especially in the case of statistically significant data described by letters.
  4. In section 2.5 from results the information in Figure 3a-c shows different results that in the text in the case of MCF-7.
  5. Also in figure 3, 4 and 5 the authors have introduced values that are not statistically significant but still there are "*" noted as the data are statistically significant. Make all the data in the Figure to be consistent, if you add values, add values to all bars and so on.
  6. In the Discussion section some phrases are not supported by the data in the figures, as I mention above due to the inconsistence of the data in figures and in the results text. 
  7. Please verify the phrases with the results in the specific Figure: "Such changes were not found when MCF-7 cells (ER-positive) were used"; "Here, however, changes were only observed in MDA-MB-231 and not in MCF-7 breast cancer cells". "significant reduction of IFN-gamma producing T cell (exclusively CD4+ lymphocytes) in response to anti-PD-1 and anti-CTLA-4 inhibition".
  8. The reference section should be uniformized, there are articles that lack , volume, issue, page no, for some article the Pubmed link is give for other there is not. Some citations are in italic other not.
  9. Also, the same experiments need to be done on at least 2 more triple negative and EP-positive cell lines in order to be able to have a correct conclusion, an experiment solely on one cell line for each group is not enough. 

Reviewer 2 Report

The manuscript: Differential response of MDA-MB-231 and MCF-7 breast can-cer cells to in vitro inhibition with CTLA-4 and PD-1 through cancer-immune cells modified interactions, poccessess several weaknesses, those might deserve the authors attantion:

  • The Introduction is not focusing on the object of the study appropriately.
  • The discussion of the used study model, methods, and obtained data about the expression of several signaling molecules is not adequate. Furthermore, the study is not sufficiently discussed in light of the current knowledge.
  • The characteristics about the cells – MDA-MB-231/MCF-7, such as a passage number, supplier of DMEM media and suitability of this particular medium for incubating the cells in atmosphere with 5 % CO2.
  • The work with PBMC, which were prepared from the blood of healthy donors, the criteria or molecular characteristics for selecting healthy blood donors are not provided as well as gender of the donors is not clearly specified. The putative effect of the applied particular batch of FBS the molecular/cellular properties of PBMC is not appropriately discussed. The details about the evaluation and characterization of PBMC are minimal. In addition, it remains unclear if PBMC isolated from several individuals were pooled or used separately in experiments.
  • For coculture experiments, the selection of 24-hour time interval is not appropriately experimentally justified.
  • The specificity of used antibodies is not experimentally justified.
  • The details about cell proliferation based on the estimation of lactate dehydrogenase activity are not provided in a sufficient manner
  • The number of repetitions/replicates is not provided.
  • The statistical analysis of the data is not sufficiently detailed. The manuscript does not contain sufficient information about the distribution test performed and the selected criterion for estimating the „non-normal“ distribution and justification of the selected non-parametric test.
  • The alternative method(s) to evaluate estimated changes in the expression levels of studied genes is (are) not performed.
  • The control columns in the graphs (Figs 1, 3 - 5 ) do not contain error bars at all, and it is not clearly communicated what the meaning of the presented error bars is. In several cases, there are untypical characters over the column in graphs, those meaning is not explained. The legends of the graphs are not sufficiently self-explanatory.

Round 2

Reviewer 1 Report

No comments.

Author Response

We are glad we managed to respond effectively to all Reviewers comments and would like to thank again for all the suggestions that contributed significantly to the manuscript improvement.

Reviewer 2 Report

The current version of the revised manuscript was modified accordingly with the majority of the suggestions.

However, several previously raised significant points have still remained untouched. These includes:

  • The discussion of the used study model and applied methods is not adequate.
  • Even authors provide the statement about the anti-CTLA-4 and anti-PD1 antibodies used, the specificity of the remaining antibodies remains unclear. Since most of the obtained results are based on the flow cytometry method, and in this respect, the specificity of the applied antibodies is one of the critical factors that can impact the recorded outcome. Therefore, it seems crucial to provide the experimental data about the specificity of antibodies or to supplement the data with the alternative method(s) to evaluate the estimated changes in the expression levels of the studied genes.

Round 3

Reviewer 2 Report

Dear Authors and Editors,

the authors responded to all my comments in appropriate way.

This manuscript is a resubmission of an earlier submission. The following is a list of the peer review reports and author responses from that submission.

Round 1

Reviewer 1 Report

The manuscript with the title “Differential response of MDA-MB-231 and MCF-7 breast cancer cells to in vitro inhibition with CTLA-4 and PD-1 through cancer-immune cells modified interactions” by Grubczak et al.  analyze the in vitro effects of inhibiting CTLA-4 and PD-1 on interactions between co-cultured lymphocytes and two selected breast adenocarcinoma cell lines, MDA-MB-231 and MCF-7.

I cannot spot which is the novelty of this presented work and, for this and other reasons, I cannot support publication of this manuscript. I’ll briefly described the reason behind this idea. Rationale of experiments is not clearly defined, results are poorly described, appropriate experimental control are missing, statistics are inappropriately applied between different cell lines and statistic value are not described or defined, cell cycle phase representation is incomplete, LDH assay is missing appropriate control, lower part of figure 2 is not described and is unclear to me what was the aims of these graphs, in fig 3 -4-5 there are repetitions in the graphs, meaning that looking, for instance, at “IFN-g+ as percentage of CD4+ lymphocytes” or at “CD4+ IFN-g+ as percentage of  lymphocytes” means look at the same cells. Similarly in fig. 4 and 5.  

The most important point is that in the majority of the experiments is not possible to state that the combination with anti-CTLA4 or anti-PD1 will induce a variation in the tumor cell-immune cells behaviour  because the same level of significance, where present, is given also by the PBMC alone added (fig. 1) or when cancer cells are “added” to combine culture (fig. 3-4-5).

Reviewer 2 Report

I have reviewed the manuscript. The authors co-cultured human breast cancer cell lines with healthy donor PBMC to study T-cell killing on tumor cells. The experimental design is problematic. There might be an MHC mismatch between tumor cell lines and PBMCs. Also, the PBMCs contain very few tumor-specific T cells. The observed effects are more likely not relevant to cancer immunity.